# AI in Dentistry: Innovations, Ethical Considerations, and Integration Barriers

**DOI:** 10.3390/bioengineering12090928

**Published:** 2025-08-29

**Authors:** Tao-Yuan Liu, Kun-Hua Lee, Arvind Mukundan, Riya Karmakar, Hardik Dhiman, Hsiang-Chen Wang

**Affiliations:** 1Department of Pediatrics, Kaohsiung Armed Forces General Hospital, No. 2, Zhongzheng 1st Rd., Lingya District, Kaohsiung City 80284, Taiwan; a1969928@gmail.com; 2Department of Medicine, National Defense Medical University, No. 161, Sec. 6, Minquan E. Rd., Neihu District, Taipei City 11490, Taiwan; 3Department of Trauma, Changhua Christian Hospital, No. 135, Nanxiao St., Changhua City 50006, Taiwan; 88847@cch.org.tw; 4Department of Mechanical Engineering, National Chung Cheng University, No. 168, University Rd., Min Hsiung, Chiayi 62102, Taiwan; arvindmukund96@gmail.com (A.M.); karmakarriya345@gmail.com (R.K.); 5Department of Biomedical Imaging, Chennai Institute of Technology, Chennai 600069, Tamil Nadu, India; 6Department of Computer Science Engineering, Chandigarh University, NH-05, Ludhiana, Highway, Chandigarh State, Mohali 140413, Punjab, India; hardik.dhiman29@gmail.com; 7Department of Medical Research, Dalin Tzu Chi Hospital, Buddhist Tzu Chi Medical Foundation, No. 2, Minsheng Road, Dalin, Chiayi 62247, Taiwan; 8Department of Technology Development, Hitspectra Intelligent Technology Co., Ltd., Kaohsiung 80661, Taiwan

**Keywords:** artificial intelligence, deep learning, cone-beam computed tomography, machine learning, Explainable AI, convolutional neural networks, robotic process automation

## Abstract

Background/Objectives: Artificial Intelligence (AI) is improving dentistry through increased accuracy in diagnostics, planning, and workflow automation. AI tools, including machine learning (ML) and deep learning (DL), are being adopted in oral medicine to improve patient care, efficiency, and lessen clinicians’ workloads. AI in dentistry, despite its use, faces an issue of acceptance, with its obstacles including ethical, legal, and technological ones. In this article, a review of current AI use in oral medicine, new technology development, and integration barriers is discussed. Methods: A narrative review of peer-reviewed articles in databases such as PubMed, Scopus, Web of Science, and Google Scholar was conducted. Peer-reviewed articles over the last decade, such as AI application in diagnostic imaging, predictive analysis, real-time documentation, and workflows automation, were examined. Besides, improvements in AI models and critical impediments such as ethical concerns and integration barriers were addressed in the review. Results: AI has exhibited strong performance in radiographic diagnostics, with high accuracy in reading cone-beam computed tomography (CBCT) scan, intraoral photographs, and radiographs. AI-facilitated predictive analysis has enhanced personalized care planning and disease avoidance, and AI-facilitated automation of workflows has maximized administrative workflows and patient record management. U-Net-based segmentation models exhibit sensitivities and specificities of approximately 93.0% and 88.0%, respectively, in identifying periapical lesions on 2D CBCT slices. TensorFlow-based workflow modules, integrated into vendor platforms such as Planmeca Romexis, can reduce the processing time of patient records by a minimum of 30 percent in standard practice. The privacy-preserving federated learning architecture has attained cross-site model consistency exceeding 90% accuracy, enabling collaborative training among diverse dentistry clinics. Explainable AI (XAI) and federated learning have enhanced AI transparency and security with technological advancement, but barriers include concerns regarding data privacy, AI bias, gaps in AI regulating, and training clinicians. Conclusions: AI is revolutionizing dentistry with enhanced diagnostic accuracy, predictive planning, and efficient administration automation. With technology developing AI software even smarter, ethics and legislation have to follow in order to allow responsible AI integration. To make AI in dental care work at its best, future research will have to prioritize AI interpretability, developing uniform protocols, and collaboration between specialties in order to allow AI’s full potential in dentistry.

## 1. Introduction

Artificial Intelligence (AI) is revolutionizing most industries including healthcare and oral medicine. AI is computer intelligence that simulates human thinking and problem-solving capabilities and forms a part of present dentistry [1]. AI’s enhanced capacity for processing big datasets, trending, and providing evidence-based information is improving disease diagnosing, planning, and work management in dentistry [2]. With the growing use of AI-facilitated diagnostics, predictive analysis, and automation, efficiency, accuracy, and care for the patient have increased in dentistry practice. AI in dentistry is in a variety of forms, such as diagnostics and planning through images, robotic surgical interventions, and personalized prosthetics [3]. AI has been successful in radiographic picture interpretation, post-treatment prognosis prediction, and prosthetic and orthodontic aligner optimization [4]. Advances in ML, such as deep learning (DL) and NLP, increasingly make AI useful for supporting dental professionals in decision-making and administration [5]. Despite such development, AI integration in dentistry is not problem-free, with its accompanying ethical, concerns regarding security over data, and uniformity requirements for protocols. There is continued activity in developing AI-enriched dental tools, with a hope for AI to enrich, not displace, expert capabilities [6,7]. As AI continues to develop, its impact on dentistry will have a deep impact, with care in dentistry becoming increasingly specific, efficient, and accessible for professionals and patients [8]. Figure 1 illustrates the diverse applications of artificial intelligence (AI) in oral medicine. It highlights how AI-powered tools enhance therapy planning, diagnostic accuracy, and workflow automation. Key technologies such as deep learning (DL) and machine learning (ML) are integrated into dentistry practices for interpreting complex radiographic images, predicting disease progression, and optimizing patient care. These advancements contribute to improved efficiency, reduced clinician workload, and personalized treatment approaches. Literature searches were performed in PubMed, Scopus, Web of Science, and Google Scholar utilizing the following fundamental Boolean operator sequence: (artificial intelligence OR AI OR machine learning OR deep learning OR neural network OR computer vision) AND (dentistry OR dental OR oral medicine OR oral health OR dental imaging). Targeted sub-searches was employed to extract application domains within specific topics by utilizing a blend of AI terminology and keywords pertinent to modalities or workflows—such as (“CBCT” OR “cone-beam computed tomography” OR radiograph* OR “intraoral photograph*”) for diagnostic imaging, and (“robotic process automation” OR RPA OR AutoML OR “workflow automation”) for workflow studies. The peer-reviewed articles were restricted to English language journals published from January 2015 to June 2025. To establish a coherent framework for our narrative review, we have identified three primary objectives:(1)to delineate the current landscape of AI integration in dental diagnostic imaging and workflow automation;(2)to identify and critically analyze significant ethical, regulatory, and technical obstacles to clinical adoption; and(3)to propose evidence-based strategies for future research, standardization, and interdisciplinary collaboration.

## 2. Current AI Applications in Oral Medicine

AI is transforming oral medicine with heightened accuracy in therapy planning, diagnostics, and workflow automation. AI technology, including DL and ML, is becoming increasingly utilized in dentistry for patient care improvement, efficiency, and manpower requirements [9]. AI technology, fueled with AI, processes complex radiography information, background information of a patient, and real-time clinic information and creates useful information for early disease prediction, personalized care, and effective administration processes [10].

The adoption of AI in oral medicine has seen improvement in diagnostics, with the prediction of disease at an early stage in oral disease through computerized radiographic analysis of radiographs. Clinicians can utilize predictive analysis for developing individualized care programs by forecasting disease progression and therapeutic efficacy. AI-powered automation is also simplifying clinic operations through less clerical workloads, augmented documentation, and enhanced communications between practitioner and patient. With continuous development in AI technology, its application in oral medicine will become ever more efficient, effective, and accessible in terms of offering alternatives for dental care [11].

### 2.1. Diagnostic Imaging and Analysis

AI has changed diagnostics in oral medicine with its ability to interpret CBCT scans, intraoral photographs, and dental radiographs accurately. AI platforms utilize DL and ML algorithms to classify and detect caries, periapical pathologies, and periodontal disease with high accuracy [12,13]. Automated programs for tooth segmentation, developed by Cui et al. [14] and Lahoud et al. [15], have been proven to have equivalent accuracy in segmentation with radiologists with a lot of experience, with considerable savings in processing time, and contribute to therapy planning. AI platforms increasingly classify abnormalities, cysts, and tumors, and in the future, AI will have a considerable role in timely and correct diagnoses [16].

Aside from its application in diagnostics, AI encourages improvement in the analysis and processing of images. Such methodologies involve AI-facilitated super-resolution (SR) processing, with a view towards CBCT resolution improvement, and allow clinicians to visualize small structures such as root canal structures, not with traditional methodologies [17,18]. Neural networks have been implemented in structures such as prosthetics, crowns, and implants, and in supporting effective diagnostics and care processes [19]. Katsumata documents the use of generative AI platforms for report simplification for diagnostics through NLP, with a view toward workflow improvement even further [20]. Diagnocat, for instance, discussed by Ezhov et al., has been proven to have high sensitivity and specificity in caries and periodontitis diagnostics, in supporting clinicians in decision processes [21].

The use of AI in CBCT imaging has changed personalized dentistry’s planning for treatment in a significant manner. CBCT-data-integrated 3D simulations with high fidelity for surgical, prosthetic, and orthodontic interventions have become a reality [22]. Automated alveolar bone and tooth segmentation algorithms, discussed in studies by Lahoud et al. and Cui et al., enable planning for interventions in a more accurate manner, minimizing both manpower and time taken for segmentation in a manual manner. AI’s application in leveraging patient-specific information, such as bone density and background of dentition, for optimizing restoration designs and implant positioning for increased patient success and efficiency in operations, is emphasized by Bonny et al. [23].

Though AI is already proving to have considerable worth, there are yet obstacles in its path. Chauhan et al., cite a need for standardized methodologies and larger, annotated datasets in an effort to mitigate variation in AI models and make them more relevant in a clinical environment [24]. Mureșanu et al., contribute that AI model external validation must take place in an effort to make them dependable in a range of clinic settings [25]. Despite obstacles, an abundance of information in these sources bears witness to AI system adaptability and longevity, with a potential for changing diagnostic imaging in dentistry. Figure 2 demonstrates the transformative role of AI in diagnostic imaging and predictive analysis within dentistry. It highlights how AI-powered tools, such as DL and ML algorithms, are utilized for accurate interpretation of CBCT scans, intraoral photographs, and dental radiographs. These technologies enable early detection of caries, periodontal diseases, and other oral pathologies. Additionally, predictive analysis facilitates personalized care planning by forecasting disease progression and treatment efficacy, contributing to improved patient outcomes and streamlined workflows. In dental diagnostic imaging, AI applications have shown significant potential and have exhibited consistent and robust clinical performance across many imaging modalities. CBCT images analyzed with deep learning algorithms demonstrate commendable accuracy in anatomical segmentation and proficient lesion characterization, aligning with clinical diagnoses. Similarly, CNN models utilized for analyzing intraoral photographs have demonstrated superior efficacy in identifying prevalent oral health conditions, including dental caries and periodontal diseases, which have been positively correlated with clinician diagnoses. Furthermore, advanced object-detection systems utilized in digital radiography analysis have streamlined and automated lesion detection processes, achieving a high degree of sensitivity and specificity, hence confirming their clinical validity and utility. Overall, these AI-based methodologies may be considered to yield significant enhancements in dentistry diagnosis, as they offer numerous benefits regarding accuracy, diagnostic performance, and the advancement of clinical decision-making. Table 1 summarizes recent applications AI in dental diagnostic imaging, highlighting the diverse methodologies, imaging modalities, and clinical objectives explored in the current literature.

Recent data indicates that CNN-based architecture remains the predominant backbone of dental imaging and planning. U-Net with improved attention mechanisms U-Net is frequently employed for precise segmentation of anatomies in CBCT and 2D radiography modalities, while Mask R-CNN and Faster R-CNN are utilized for contemporaneous detection and segmentation of lesions. The majority of researchers employ ResNet-based classifiers and VGG16 networks due to their proficiency in extracting deep features for the analysis of intraoral pictures and lesion classification. YOLOv5 and YOLOv8 real-time detection frameworks have been developed and are recognized for their rapid and precise identification of periapical and interproximal lesions. Transformer models have recently appeared, potentially offering enhanced contextual comprehension; nevertheless, they are not yet as prevalent as present CNN methodologies.

### 2.2. Predictive Analytics for Treatment Planning

Predictive analysis is improving treatment planning in dentistry with AI, ML, and big data analysis in a quest for optimized decision-making and patient success. AI-powered predictive models scan patient files, radiography, and demographics to make predictions regarding future oral disease, individualized care, and optimized clinic efficiency [26]. By predicting cases at high risk for caries, periodontal disease, and malignancies, predictive analysis can allow for early intervention and preventive care, in an attempt to save overall healthcare dollars and enhance patient wellness.

AI-facilitated predictive analysis is increasingly being adopted in planning for dental care, providing computerized feedback regarding disease progression and best-fit patient-specific protocols for care. Methods in deep learning, such as CNNs and recurrent neural networks (RNNs), enable high accuracy in predicting care outcomes [27]. The algorithms enable deciding best-fit modalities for prosthetic rehabilitation, orthodontic realignment, and surgical intervention, with a patient’s individualistic radiographic and clinical information in view [28]. Aside from that, predictive analysis with ML is an important function in practice management in dentistry, with optimized scheduling, inventories, and forecasting for increased operational efficiency [29]. All such implementations simplify workflows and reduce workloads for professionals and dentists; but with increased patient care.

Tele-dentistry, when paired with predictive analysis, creates avenues for personalized care through distance consultation and planning [30]. AI-powered decision-support tools decode real-time patient information, and dentistry professionals can make care-related decisions with assurance. Simultaneously, early disease diagnoses through faint trends in patient information not perceivable with traditional diagnostics become a reality through algorithms in machines [31]. AI-powered planning even reaches surgical interventions, such as mandibular third molar extraction, in which predictive algorithms utilize CBCT for extraction difficulty analysis and complications prediction, improving accuracy in operations and patient security [32].

Another important area that leverages predictive analytics is personalized care planning for dentistry. AI algorithms review disparate sources of information, including genetic profiles, medical histories, and radiologic studies, and utilize them to make personalized care recommendations [33]. Personalized care planning enables precision dentistry, with care protocols individualized for each patient’s specific needs, with reduced overreliance on protocols for a general case. AI algorithms for optimization extend even deeper and virtualize and make care protocols adaptable through simulation and long-term forecasting [34]. Business analysis and predictive modeling are utilized in overcoming inequity in access to care through predictive modeling and effective distribution of assets [35].

Artificial intelligence planning for therapy enables processes to become efficient through the integration of AI in workflows in routine work in dentistry. AI-powered diagnostics contribute towards disease severity analysis and therapy options, with a reduced cognitive burden for professionals in dentistry [36]. Precision dentistry reaches new horizons through predictive modeling, with an improvement in estimation and stratification of risk, and preventive intervention [37]. Decision support through AI-powered tools aids with increased patient involvement by providing interactive platforms for patient information regarding options for therapy and prognosis [38].

Deep learning has played an important role in transforming treatment planning with enhanced accuracy in diagnosing dental disease and maximizing restoration, orthodontic, and prosthetic processes. AI algorithms in endodontics predict root canal therapy success, and in surgical and prosthetic dentistry, AI optimizes surgical planning with precisely designed prosthetics and implant positioning [39]. Apart from such big data analysis, neural network model optimization is being applied for early disease prediction and planning, and infection and fatigability prediction [40]. AI and predictive analysis have touched dentistry, changing our care planning with evidenced, digitalized, and patient-centered care. With its progression, such technology will become even more efficient in defining the role of dental practice with accuracy and efficiency and in improving patient well-being. Figure 3 illustrates the integration of AI-driven automation in dental practice workflows. It showcases how robotic process automation (RPA) and AI-powered systems streamline administrative tasks, such as patient record management, appointment scheduling, and real-time documentation. By reducing clerical workloads, these technologies enhance communication between practitioners and patients, improve clinic efficiency, and allow dental professionals to focus on providing high-quality patient care. The figure emphasizes the role of AI in transforming traditional workflows into more efficient and scalable processes.

Several predictive models have been developed, validated, and therapeutically implemented. Accurate risk-assessment tools for periodontal disease can incorporate patient-specific variables, such as age, health history, and oral hygiene, to effectively forecast disease progression, thereby facilitating the customization of preventive care programs in routine dental practice. Similarly, predictive analytics in orthodontics can assist physicians by forecasting treatment durations and outcomes based on patient variables, so enhancing decision-making and optimizing treatment planning. Other potentially useful predictive analytics techniques remain in the development and validation phase and have not yet been widely applied. These are sophisticated models designed to forecast implant success utilizing complex machine learning algorithms trained on extensive historical datasets, with AI-driven predictions of oral cancer growth based on genomic and imaging indicators. Despite the preliminary outcomes of these research indicating significant potential, extensive clinical validation and approval remain necessary before full-scale deployment may occur. Table 2 shows the summary of Predictive Analytics in Dentistry in terms of clinical applications as well as theoretical applications.

### 2.3. AI in Real-Time Documentation and Workflow Automation

The application of AI in real-time documentation and workflows automation is revolutionizing dental care through efficiency, lowered administration workloads, and heightened patient engagement. AI-facilitated automation reduces processes such as documentation, care workflows, and patient record management, allowing for less work for dental professionals and allowing them to spend more time with patients. AI-facilitated tools have enhanced accuracy and efficiency in reporting, scheduling, and data entry, and an optimized and efficient clinical environment [41,42].

AI has seen an era of data dentistry with electronic health records (EHRs) supplemented with AI analysis. Automated machine learning (AutoML) platforms have started to sort and classify tremendous volumes of dental information, reducing documentation work significantly [43]. Data dentistry, Schwendicke et al., say, integrates a range of sources such as demographics, social, clinic, and environment, and forms a real-time, overall analysis of a patient’s state of health [44]. With predictive analysis and decision-making, overall care can become a success. Besides, AI technology in patient information and communications encourages increased patient involvement through interactive platforms for patient inquiry, virtual consultation, language translation software, and proper communications between dentists and patients [45].

AI big-data technology is transforming administration processes in dental clinics, too. Computerized patient files and AI analysis make laboratory and clinic processes easier, such as computerized impressions and CAD-CAM technology, and virtual simulation of a patient [46]. RPA is increasingly becoming a prevalent technology for clerical work processing, such as billing, scheduling, and claims processing, and for reducing clerical errors and operational effectiveness [47]. AI integration in clinic processes allows for continuous toggling between care phases, such as diagnosis and execution, through analysis, documentation, and automation of errors [48].

The use of AI and robotic technology in autonomous dental care platforms has taken the automation of processes to an even new level. AI-enforced RPA roboticizes repetitive, rule-bound processes such as documentation processing, allowing for free time for dental professionals to attend to patient care [49].

In administration in the healthcare sector, automation works best, with AI-powered scheduling, predictive analysis, and workflow maximization enhancing resource use and cutting down expenses [50]. AI-facilitated laboratory automation platforms have also become increasingly popular, especially in times of the COVID-19 pandemic, allowing for remote operations, maximization of task management, and reduced intervention in high-volume processing [51].

Moreover, AI-facilitated workflow automation helped in processing unstructured documents. Traditional documentation processes have a problem with disparate structures and forms, and, as such, information extraction is cumbersome and susceptible to errors. AI-facilitated techniques today deliver better information retrieval and processing of unstructured clinic observations, and efficiency in medical recordkeeping is boosted [52,53]. AI claims processing software maximizes workflows through reduced repetitions and inaccuracies in medical reimbursement, timely processing, and reduced administration blocks [54]. AI has also played a part in clinic work management, with its contribution towards decision-making and work efficiency through repetitive work automation, predictive analysis improvement, and efficient group collaboration [55]. AI reporting tools simplify documentation in testing automation, with actionable information for effective decision-making and patient care [56]. Not only will such improvement make operations efficient but in patient care through real-time tracking and analysis of clinic information. AI in workflow automation in dentistry bears numerous positive impacts, but its concerns over data privacy, ethics, and compliance with laws and legislation must first be addressed in a manner that will enable responsible implementations of AI. Despite that, with continued AI technology development, its integration in real-time documentation and workflow automation will go a long way in enhancing efficiency, accuracy, and patient care in dentistry. Figure 4 highlights the advancements in AI transparency and security through Explainable AI (XAI) and federated learning technologies. XAI enhances the interpretability of AI models, allowing dental professionals to better understand the reasoning behind AI-driven decisions, thereby improving trust and adoption. Federated learning addresses data privacy concerns by enabling decentralized training of AI models across multiple institutions without sharing sensitive patient data. Together, these technologies pave the way for ethical and secure AI integration in dentistry, ensuring both accuracy and accountability in clinical applications. Recent advancements indicate that hybrid models can enhance dental applications by integrating imaging features derived from CNN-based architectures with patient-level clinical metadata, such as age, smoking status, periodontal measurements, and medical history, thereby surpassing purely image-based networks. Integrating morphological data with individual risk factors enhances overall accuracy and decreases false-positive rates in these multimodal systems. Moreover, information integration enhances model robustness across diverse patient cohorts and enables tailored risk classification, hence aiding treatment planning and acting as a bias-reducing attribute of solely image-based methodologies. Table 3 shows the summary of ai tools and applications in real-time documentation and work-flow automation.

## 3. Technological Advancements in AI Models

The rapid development in AI revolutionized dentistry practice, with advanced models designed to maximize diagnostics, planning, patient care, and clinic workflows [57]. AI technology in deep learning, machine algorithms, and natural language processing (NLP) has revolutionized dental care through predictive capabilities, efficiency, and accuracy [58]. Technological development facilitated AI to move from a hardware orientation towards a software orientation, with a heightened boost in automation in educational and oral care setting [59,60].

One of the most important AI breakthroughs in dentistry is its application in computerized interpretation of dental images [10]. Traditional diagnostics rely on manual analysis, and such analysis can both be incorrect and tedious. AI algorithms have increased accuracy in diagnostics via X-ray, near-infrared, and CBCT scan analysis with high accuracy [61]. AI algorithms apply CNNs for caries, periodontal disease, and periapical lesion detection and caries, periodontal disease, and periapical lesion diagnoses, and for early intervention and efficient planning for intervention. AI-implemented implant prognosis algorithms have also exhibited high accuracy, having over 99% accuracy in implant success and failure prediction via radiographic analysis [62].

Recent developments in AI integration have even extended to intelligence augmentation (IA), whose function is to enhance decision-making in humans and not replace them. AI algorithms today assist dentists with real-time recommendations and predictive analysis, supporting individualized care and success in procedures [63,64]. Besides, big language models (LLMs) such as ChatGPT 5 have changed dental diagnostics through supporting doctor-patient conversation and computerized documentation processes [58]. AI-powered virtual assistants ease patient encounters, enable diagnostic inquiry, and enable patient activation through amplified personalized and sensitive channels of conversation. Figure 5 outlines the primary ethical, legal, and technological challenges faced in integrating AI into dentistry. Key issues include concerns over data privacy, algorithmic bias, lack of standardized protocols, and insufficient training for clinicians. The figure emphasizes the importance of developing XAI models, federated learning techniques, and collaborative frameworks to address these barriers. It also highlights the need for robust legislation and ethical guidelines to ensure responsible and effective AI adoption in dental practices.

AI is transforming dental training with smart tutor platforms and real-time feedback for clinical practice. Multimodal base models, with a mix of text, picture, and voice information, have supported virtual simulations and case studies for hands-on practice for students [65]. AI integration in training in dentistry helps students receive personalized assessments, with a boost in technical skills and clinic practice preparedness. Nevertheless, concerns regarding data privacy, AI training bias, and regulative controls have to be addressed for the responsible and ethical integration of AI [66].

Beyond clinical diagnostics and training, AI is transforming numerous specialties in dentistry with precision dentistry. AI algorithms are increasingly being adopted in orthodontics for computerized cephalometric analysis, in prosthodontics for personalized prosthetics, and in periodontics for disease progression tracking [67]. Nanorobotic dentistry, a technology in development, is under investigation, with studies looking at how AI-powered nanobots can make less invasive interventions in dentistry and target specific drug delivery [68].

Another key advancement in AI models is in developing XAI and federated learning. In contrast to traditional AI models, whose performance in a “black box” with little interpretability, XAI seeks to make AI decision-making processes transparent [69] and understandable for clinicians. Federated learning, on its part, helps AI models learn in a decentralized manner, with anonymity for patients and with a heightened model accuracy [70]. All these breakthroughs instill confidence in AI use and allow for increased use in a clinical environment.

Despite these breakthroughs, AI in dentistry is not yet free of complications. Most AI algorithms deal with big, labeled datasets in an efficient manner, but scarcity and unbalanced datasets become a real challenge [71]. Regulatory and ethical concerns regarding AI use in practice must, therefore, be addressed in a manner that adheres to legal frameworks and protocols for patient security [72]. In the future, dentistry increasingly opens its doors to new and emerging technology with AI, such as neuroevolutionary and quantum AI, holding hope for increased success in dentistry diagnostics and therapy [73]. AI’s increased contribution towards future dentistry re-emphasizes the imperative for continued studies, multidisciplinary, and responsible AI integration in a quest for utmost gain and a counterbalance for any loss [74]. With such new technological breakthroughs, AI will redefine dentistry, driving innovation, accuracy, and efficiency in clinic and educational environments.

## 4. Challenges and Limitations

Despite its transformational potential, AI integration in dentistry is challenged with a variety of obstacles. Patient information privacy, consent, and accountability for AI-made recommendations rank high in terms of ongoing ethical and legal concerns. Regulatory structures for AI use in medical practice have not yet matured, and therefore, uncertainty about compliance and accountability prevails. There are technological and financial barriers to AI integration, such as high costs for AI-powered diagnostics and poor interoperability with existing dental software. Inadequate training in AI use for most dental professionals and hesitation in its use contributes to its lack of integration. High costs for AI-powered diagnostics and poor interoperability with existing dental software make integration even more challenging. All these barriers can only be addressed through collaboration between regulators, researchers, and dentists. Regulatory policies, AI training, and technological development will make responsible and effective use of AI in dentistry a reality. What follows below are specific ethical, legal, and technological barriers that will have to be addressed for AI to become its best in current dental practice. Figure 6 illustrates the key challenges and limitations faced during the integration of AI technologies into dentistry. It highlights critical aspects such as data privacy concerns, algorithmic bias, lack of standardized protocols, and insufficient training for clinicians. Ethical considerations are emphasized, including the need for transparency, accountability, and equitable access to AI-driven tools. The diagram also underscores the importance of collaboration between dental professionals, AI developers, and policymakers to overcome these barriers and ensure responsible AI adoption.

### 4.1. Ethical and Legal Concerns

The integration of AI in dentistry is accompanied with a lot of ethical and legal concerns, and for that, wise laws and ethical frameworks must dominate. With AI technology becoming a reality, concerns regarding patient information confidentiality, consent, accountability, and independence in decision-making have become a matter of greatest concern in ethical consideration [75]. With the increased use of AI technology in dentistry, developing in-depth legal frameworks for protecting patient rights, transparency, and minimizing ethical concerns in AI-facilitated decision-making in dentistry is an issue of greatest concern.

One of the most important concerns about AI in dentistry is protecting patient information security and confidentiality. AI technology handles massive amounts of patient information, including radiographic files, medical histories, and real-time diagnostics, and raises concerns about unauthorized access, loss, and misuse [76]. AI use must entail offering full information about the collection, storage, and use of information to patients. Patient consent must, therefore, be acquired for the use of AI-powered decision-making in diagnosing and planning therapy [77]. Nevertheless, a lack of harmonized legislation about AI processing of information presents enormous barriers to imposing protective factors for information in a range of settings [78].

Another ethical issue in AI dentistry is bias and fairness in AI algorithms. AI algorithms learn with big datasets, but when datasets lack diversity, AI algorithms can exaggerate biases and yield inequalities in dental planning and diagnoses [79]. AI frameworks have to prioritize diversity, equity, and inclusion for AI tools to produce fair and correct information for a range of patient groups [80]. AI decision-making must, in addition, be developed in a manner that complements, and not substitutes, expert professionals’ capabilities, with clinicians having latitude for exercising critical thinking and independence in working with patients [81]. The legal dimensions of AI in dentistry, in fact, extend even to accountability and liability, too. Where AI programs make incorrect and damaging diagnoses, it is not even assured whose accountability will fall, a dentist, AI developer, or institution employing technology [82]. Present laws cannot cover AI-dependent medical decision-making, and new legal frameworks must prescribe accountability in AI-facilitated medical practice. There is an ethical issue, in fact, in not employing AI for cost-saving at the expense of patient care, with financial incentives possibly having an impact in AI development and application in therapy planning [83]. Besides, ethical use of AI in training must then specifically be addressed. AI and robotic use in training in dentistry have increased diagnostics training and simulation in therapy, but then ethical concerns must then then then be addressed in providing access to AI-facilitated training tools [84]. Concerns about bias in AI educational models must be addressed, and students must be trained to employ AI responsibly. With increased use of AI in dentistry, having worldwide ethical requirements and legal frameworks in position to enable accountability, patient protection, and responsible use of AI is critical. As AI holds a high level of potential in changing dental care, its use must comply with ethical values such as transparency, patient autonomy, and fairness in a manner that will preserve both professional integrity and patient trust. There is a necessity for ongoing discussion between and amongst stakeholders, including policymakers, dentists, and AI engineers, in developing an ethical and legally compliant environment for AI-powered dentistry.

To contextualize the ethical and legal landscape of AI in dentistry, it is essential to acknowledge potential regulatory modifications underway. The recently proposed European Union AI Act represents a significant global effort to regulate AI systems through legal frameworks, establishing risk categories and corresponding mandates for transparency, robustness, and data protection [85]. This initiative will emphasize ethical accountability and risk management to ensure the ethical application of AI in healthcare and dentistry. Similarly, the United States Food and Drug Administration (FDA) is exemplary and offers explicit rules recognized as the Software as a Medical Device (SaMD) regulatory framework [86]. These guidelines establish rigorous performance assessment criteria, explicit risk assessment benchmarks, and protocols for clinical validation, pre-market approval, and post-market surveillance. Compliance with FDA criteria renders AI-assisted dental technologies subject to stringent safety, efficacy, and ethical standards, significantly impacting their acceptance and application in clinical practice [59]. The incorporation of regulatory frameworks demonstrates ongoing efforts to ensure responsible, safe, and advantageous AI integration in dentistry, highlighting the importance of compliance and governance in knowledge development that will facilitate AI use in clinical practice.

### 4.2. Integration Barriers

The integration of AI in dentistry, with its potential, is not free from several obstacles that make its general use challenging. All such obstacles occur through technological, financial, regulative, and educational factors for both dentists and patients. For AI to become an integral part of workflows in dentistry, overcoming such obstacles will make its integration and full use in an effective manner a reality.

One of the greatest impediments in AI application in dentistry is professionals’ unfamiliarity and technical expertise in AI application. Practitioners lack awareness and familiarity with AI tools and, therefore, disbelief and reluctance in utilizing AI-powered diagnosing tools [87]. In a survey conducted in Croatia, 71% of dentists agreed with AI’s usability, but 76% of them have not utilized AI tools in practice, citing a high awareness-applicability barrier for it. Besides, most respondents rated AI awareness as poor and fair, citing a need for additional training and educational programs for professionals with capabilities for effective integration of AI tools. Similarly, Singh et al., determined that students and professionals in India agreed with AI’s usability in planning and diagnosing but ranked lack of training and lack of technical infrastructure amongst principal impediments in its use [88].

Another major issue is its high integration cost. AI technology in dentistry involves high investments in infrastructure, software, and hardware, and most dental clinics, including small ones, cannot access them [89]. Procurement and maintenance of AI tools, in addition to training and software update costs, present a high barrier for integration. Cost burden is added through uncertainty in return on investment, with most clinics not wishing to make investments in AI in the absence of concrete assurance of long-term cost-effectiveness [87]. In addition to budget constraints, integration complications in workflows pose significant barriers. AI platforms must have flawless compatibility with existing dental software, such as EHRs and computerized imaging software, in a position to function effectively. Most AI algorithms, however, have not yet achieved full interoperability with traditional dental practice management software, and integration, therefore, is challenging and takes long [90]. Dental professionals face difficulty in altering workflows in a manner that incorporates AI-powered diagnostics and planning tools, and slow acceptance is a result. Integration complications through lack of uniform protocols for integration, with disparate systems having disparate integration methodologies, contribute to added complexity [91]. Trust and liability concerns also cause hesitation in AI use in dentists and dental professionals. Practitioners have concerns regarding AI accuracy in diagnoses and planning, with AI-predicted diagnoses conflicting with clinicians’ expertise [89]. There is no accountability resolution in case of wrong diagnoses facilitated through AI, with no one knowing whether it is the dentist, AI system developer, or clinic utilizing technology [92]. There is no regulating mechanism for accountability, and such hesitation in AI use is a contributing cause. Alongside algorithmic advancements, the effective implementation of AI in standard dentistry practice necessitates stringent software certification and flawless hardware integration. Dental AI solutions must traverse regulatory frameworks, including FDA 510(k) or De Novo certifications in the United States and CE marking under the European Medical Device Regulation, to establish safety, efficacy, and risk management in compliance with SaMD recommendations. Simultaneously, numerous equipment manufacturers are integrating learned AI models directly into CBCT scanners, intraoral cameras, and chairside consoles via strategic alliances with software developers, facilitating real-time image analysis without interrupting existing procedures. Ultimately, continuous post-market validation—encompassing prospective clinical trials, real-world evidence gathering, and adherence to ISO/IEC 62304 medical device software lifecycle standards—guarantees ongoing performance assessment, prompt updates, and enduring clinician confidence, all of which are critical for broad acceptance in clinical practice.

From a patient’s perspective, concerns about security and confidentiality of information discourage trust in AI technology in dentistry. AI technology requires large datasets for training and decision-support, and for that reason concerns about misuse and unauthorized access to sensitive patient information [76]. Patients will not submit private medical information when they have no information about storing, processing, and protecting it. Mitigation of such concerns through effective data protection policies and transparent patient consent processes is critical for trust establishment in AI technology in dentistry. Furthermore, job displacement anxiety for supporting and dental technicians creates a challenge in addition to such concerns. AI and machine learning technology increasingly automatizes prosthetic planning, radiographic evaluation, and planning for care, and creates concerns regarding a reduced demand for human skill [90]. AI, even when programmed to work in an auxiliary role and not a professionals’ substitution, creates a concern in many technicians’ minds that increased automation will make them lose jobs in dental labs and clinics. Adopting a participatory model in which AI works in an auxiliary role and not a professionals’ substitution can reverse such concerns [93]. Despite such obstacles, AI can have a significant impact in transforming dental care through increased accuracy in diagnostics, optimized planning for care, and increased efficiency in workflows. To make integration a reality, such obstacles will have to be eliminated through focused educational programs, financial incentives, legislation development, and planning structures for effective integration. As AI technology continues to develop, collaboration between AI developers, researchers, policymakers, and dentists will become increasingly critical in overcoming integration obstacles and in utilizing AI responsibly and effectively in dentistry.

Figure 7 outlines key strategies for advancing AI integration in dentistry to unlock its full potential. It emphasizes the importance of developing standardized protocols for AI applications, fostering interdisciplinary collaboration between dental professionals, AI developers, and policymakers, and prioritizing research on AI interpretability and transparency. Additionally, ethical considerations such as equitable access to AI technologies, data security, and minimizing algorithmic bias are highlighted as critical areas for future exploration. These initiatives aim to ensure responsible and effective implementation of AI in dental care, ultimately enhancing patient outcomes and professional efficiency. The predominant reasons for failure in actual dental AI implementations can be attributed to limitations on both the development and clinical practice sides, as discrepancies in these areas may lead to model degradation when applied to varying imaging protocols, hardware, and patient demographics. The issue lies in workflow integration, since a disorganized interface or inflexible software-hardware interface may disrupt physicians’ routines and deter uptake. Blindly adhering to AI recommendations without adequate oversight may perpetuate errors, while excessive false alerts might result in alert fatigue. Furthermore, insufficient user training regarding AI capabilities and limitations frequently results in misuse or misinterpretation of the outcomes. Finally, data breaches pose legal and trust issues, necessitating robust governance and clinical education on the implementation of AI in dentistry practices.

## 5. Future Perspective and Research Directions

The future of AI in dentistry holds tremendous potential for transforming diagnostics, planning, and workflows automation, and for a more efficient and patient-centered delivery of care for the mouth. As AI algorithms mature, integration with emerging technology such as federated learning, XAI, and quantum processing will allow new breakthroughs in care in dentistry through heightened accuracy, transparency, and security of information. AI diagnostics will become even more accurate and capable of early disease prediction, and clinicians will have a chance to act early and reverse disease progression to a later stage. AI-designed personalized planning through genetic, radiographic, and clinical information will allow for precision dentistry with personalized care for individual cases.

One of the most important avenues for future research is overcoming bias in AI, diversity in datasets, and external model testing for predictive algorithms. AI algorithms must be trained with larger and more heterogeneous datasets in an attempt to make bias-free and fair decisions in a range of populations. There must be future work in harmonization of AI application in dentistry through establishment of frameworks for AI application, including protocols for AI-guided decision, accountability, and ethics. There is an important role for multidisciplinary collaboration between AI engineers, clinicians in dentistry, and policymakers in delivering responsible and transparent AI application in a clinic setting.

The expansion of AI in robotic dentistry is yet another path with a high level of potentiality. Robotic machines with AI will soon contribute towards accuracy in surgical procedures, prosthetic planning, and orthodontic treatment, minimizing workloads and clinic efficiency. Besides, a mix of AI with augmented reality (AR) and virtual reality (VR) could revolutionize dental training, providing real-time simulations and hands-on training for dental students.

## 6. Conclusions

The integration of AI in dentistry has changed diagnostics, planning, and workflows automation, with enhanced efficiency, accuracy, and patient care orientation. AI tools including DL and ML algorithms have accelerated disease prediction, predictive analysis, and personalized modalities of care. AI documentation and AI-guided efficiency in workflows have maximized administration, reduced workloads, and relieved professionals for increased patient care. In its development, AI in dentistry continues to face impediments in terms of ethical, legal, and integration-related concerns, including bias, high cost, and incompatibility with today’s technology, for its general acceptance for use. All these have to be addressed for its full acceptance in practice. Trends including XAI, federated learning, and robotic interventions have a chance to bridge these impediments and expand its impact in dentistry even further. As AI continues to unfold, clinicians, researchers, and policymakers will have to increasingly work together in providing responsible and effective integration. With overcoming current constraints and improvement in AI algorithms, dentistry can best capitalize in AI’s potential in enhancing clinical performance, enhancing patient experiences, and developing new standards in oral care. Numerous collaborative initiatives are currently in progress to standardize dental artificial intelligence. The FDI World Dental Federation’s AI Task Team is formulating consensus guidelines, whereas the ADA’s AI Working Group is establishing best-practice standards and permission forms. Furthermore, the RSNA Dental AI Challenges and the ISO updates from CEN/TC 106 serve as benchmarking instruments and provide AI-specific standards for dental device software. The future scope involves creating and compiling an open-access, annotated dental pictures and electronic health record datasets, accompanied by a standardized set of benchmarking protocols, to enable a transparent assessment of novel AI systems within the organizations. Executing multi-center prospective randomized controlled studies to thoroughly assess the safety, efficacy, and applicability of AI tools in dentistry offices across diverse patient populations. Evaluating and deploying integrated decision-support systems that utilize AI predictions in conjunction with physician experience to assess their impact on diagnostic accuracy, workflow efficiency, and patient happiness, ensuring seamless integration into standard practice.

## Figures and Tables

**Figure 1 bioengineering-12-00928-f001:**
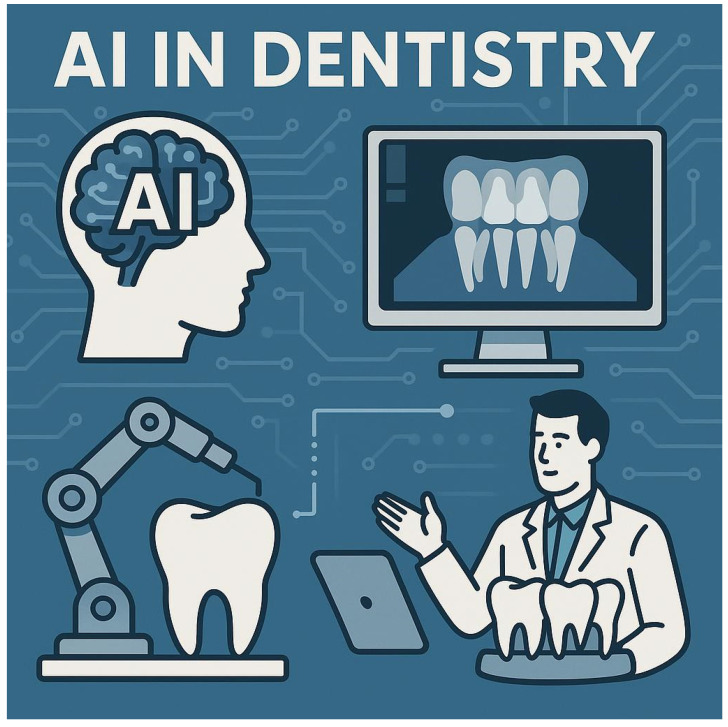
Current AI Applications in Oral Medicine.

**Figure 2 bioengineering-12-00928-f002:**
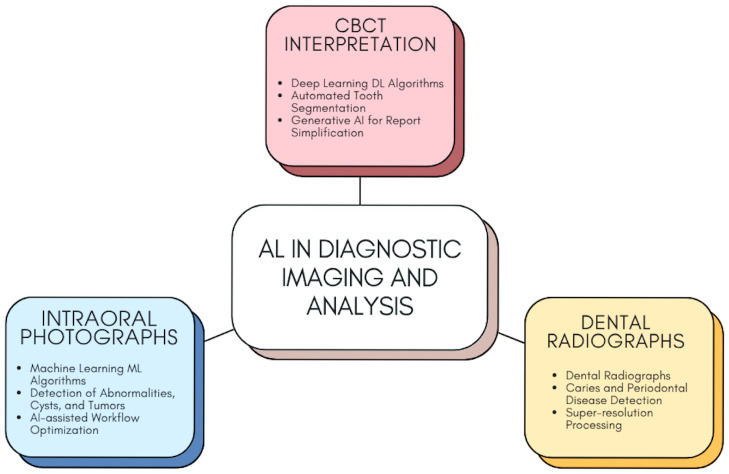
AI in Diagnostic Imaging and Analysis.

**Figure 3 bioengineering-12-00928-f003:**
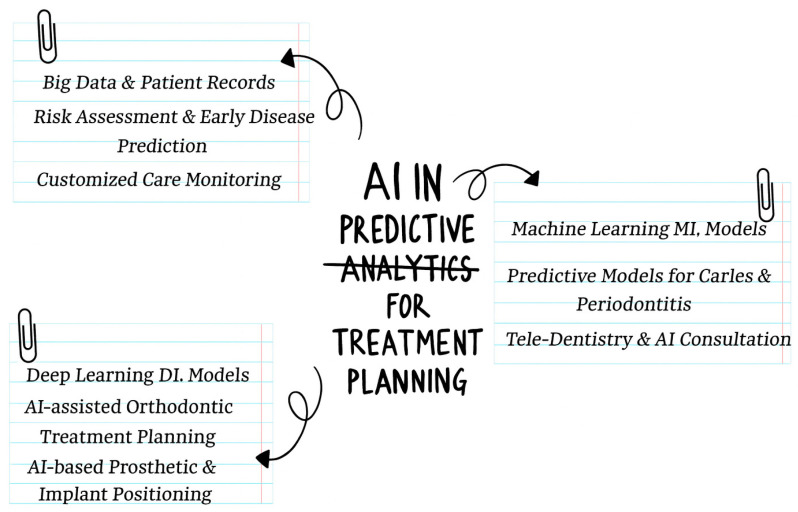
AI in Predictive Analytics for Treatment Planning.

**Figure 4 bioengineering-12-00928-f004:**
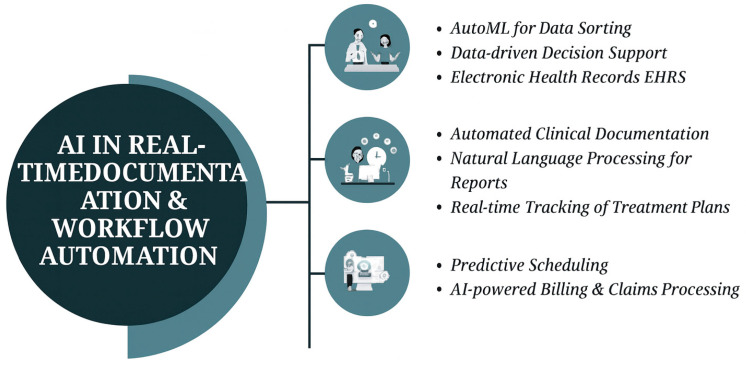
AI in Real-Time Documentation and Workflow Automation.

**Figure 5 bioengineering-12-00928-f005:**
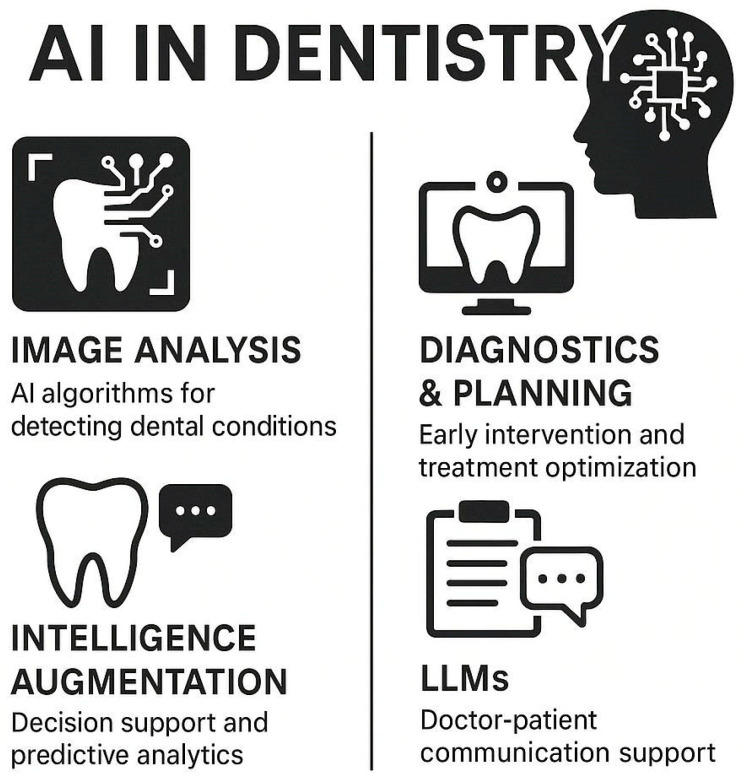
Ethical Challenges and Integration Barriers in AI-Powered Dentistry.

**Figure 6 bioengineering-12-00928-f006:**
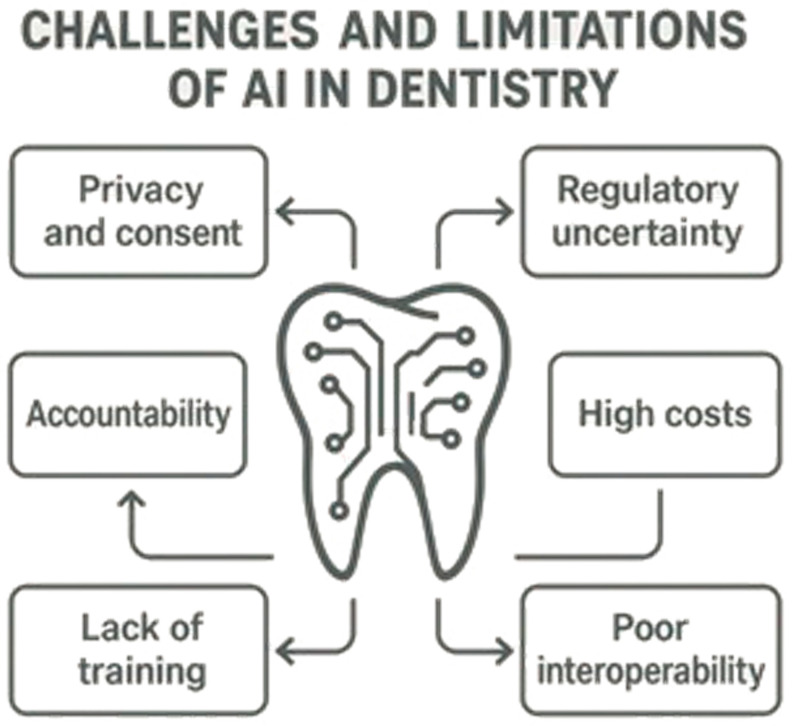
Challenges and Limitations of AI in Dentistry.

**Figure 7 bioengineering-12-00928-f007:**
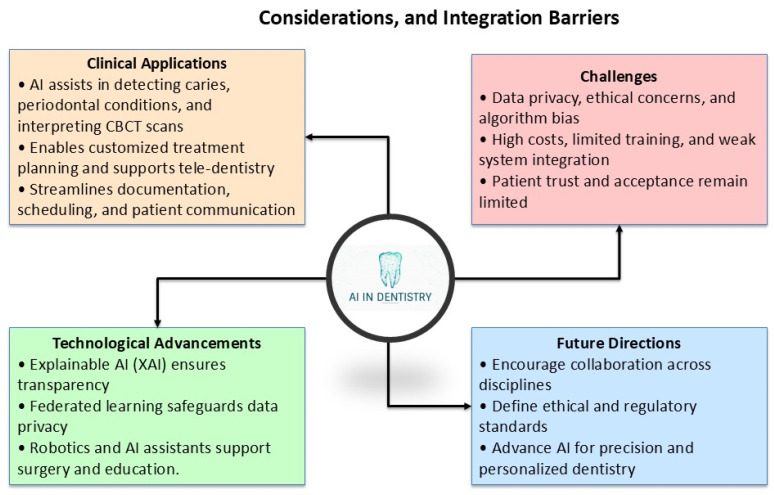
Future Directions for AI Integration in Dentistry.

**Table 1 bioengineering-12-00928-t001:** Summary of AI Applications in Dental Diagnostic Imaging.

Imaging Modality	Diagnostic Task	Common AI Models	Performance Metrics (Generalized)	Clinical Validation Level
Cone-Beam Computed Tomography (CBCT)	Anatomical segmentation, lesion detection	U-Net, Mask R-CNN, ResNet-based CNNs	High accuracy (>90%), strong lesion detection (AUC ~0.90)	Validated via prospective clinical and multi-center studies
Intraoral Photography	Caries detection, periodontal classification	VGG16, ResNet50, DenseNet	Accuracy (~90–95%), high clinician correlation	Pilot studies, clinician correlation studies
Digital Radiography	Periapical lesion identification	YOLOv8, Faster R-CNN	High sensitivity (~90%), specificity (~85–90%)	Retrospective dataset validations across multiple clinics

**Table 2 bioengineering-12-00928-t002:** Summary of Predictive Analytics in Dentistry: Clinical vs. Theoretical Applications.

Predictive Application	Clinical Status	Example Use-Cases	Validation and Impact
Periodontal Disease Progression	Clinically Implemented	Routine risk assessments, preventive care	Validated via longitudinal clinical trials; widely adopted
Orthodontic Treatment Outcomes	Clinically Implemented	Treatment duration, personalized planning	Prospective studies confirming predictive accuracy in clinics
Implant Success Prediction	Emerging/Theoretical	Implant planning, personalized risk analysis	Early-stage retrospective analyses; ongoing clinical validation needed
Oral Cancer Prognostics	Emerging/Theoretical	Cancer progression prediction, precision care	Promising preliminary results; awaiting larger clinical trials

**Table 3 bioengineering-12-00928-t003:** Summary of AI Tools and Applications in Real-Time Documentation and Workflow Automation.

AI Tool/Technology	Key Applications	Clinical Benefits
Automated Machine Learning (AutoML)	Sorting and classifying dental data (EHRs)	Reduced documentation workload, improved data management
Robotic Process Automation (RPA)	Billing, scheduling, claims processing	Reduced clerical errors, improved operational effectiveness
AI-enhanced Patient Communication Tools	Interactive patient inquiry, virtual consultations, language translation	Enhanced patient engagement and satisfaction
CAD-CAM and Virtual Simulations	Computerized impressions, treatment planning	Streamlined clinical and laboratory workflows
AI-powered Scheduling & Predictive Analysis	Appointment optimization, resource allocation	Enhanced resource use, reduced administrative costs
Laboratory Automation Platforms	Remote lab operations, task management	Improved productivity, minimized manual intervention
AI Document Processing	Processing unstructured documentation, recordkeeping	Increased efficiency, improved information retrieval
AI Claims Processing Software	Reimbursement management	Reduced errors and delays, faster claims processing
XAI	Improved transparency and interpretability of AI decisions	Enhanced clinician trust and adoption
Federated Learning	Decentralized AI model training, data privacy preservation	Improved data security, ethical AI integration

## Data Availability

The data presented in this study are available upon considerable request to the corresponding author (H.-C.W.).

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
