# Peer review of "AI in Dentistry: Innovations, Ethical Considerations, and Integration Barriers"

_bioengineering, 2025, doi:10.3390/bioengineering12090928_

Round 1

Reviewer 1 Report

Comments and Suggestions for Authors

The authors presented a work on the title "AI in Dentistry: Innovations, Ethical Considerations, and Integration Barriers" The use of non-degradable polymer (polyetheretherketone) 2 in personalized orthopedics – Review article" for the MDPI Bioengineering journal. 
However I feel authors can make following corrections to improve the review quality I recommend for minor revisions 

1. Which keywords or Boolean logic strings were employed in the database searches?
 2. Was the PRISMA framework adhered to, and if so, what is the rationale for the absence of a flow diagram?
 3. What is the final count of selected articles, and which quality assessment tool was employed for their evaluation?
 4. It would be beneficial for the authors to present a table summarizing the performance metrics of various AI models (e.g., CNN, RNN, transformer-based models) across different diagnostic modalities.
 5. In the absence of meta-analysis, what criteria were used to assess the robustness of the evidence?
 6. Which AI algorithms, such as ResNet, YOLO, U-Net, and transformers, have been most commonly cited in dental imaging or planning applications?
 7. Have hybrid models, such as CNNs combined with clinical metadata, been examined in the referenced literature?  What were their relative advantages?
 The manuscript notes "ethical and legal" challenges but fails to include citations from regulatory authorities such as the FDA, EMA, or local dental councils.
 9. What common failure modes of AI deployment in real-world dental clinics have been identified in the literature?
 The conclusion highlights the necessity for protocol development and interdisciplinary collaboration.  Are there particular examples or consortia, such as OpenMMLab Dental or the DentalXAI Consortium, that are actively engaged in these initiatives?

Reviewer 2 Report

Comments and Suggestions for Authors

The title of this paper is very interesting. The aim of this paper, as stated in the abstract, is to conduct a systematic review. The methodology and structure of this paper do not follow the aforementioned aim. In particular, authors should have followed the PRISMA statement methodology. Based on the above, the conclusions of this study could not be considered reliable. My suggestion to the authors' team is to rewrite the paper following the proper methodology of a systematic review.

Reviewer 3 Report

Comments and Suggestions for Authors

The manuscript presents a review of the contemporary developments in the area of artificial intelligence and machine learning for dentistry. There are 90 references from 2018 to 2025. A lot of important aspects of automation with AI and ML in dentistry are discussed.

As the manuscript is positioned as the systematic review the paper selection criteria is to be provided. The statistics on the number of documents available in selected databases and the number selected is to be presented. The dynamics in time and the geographic distribution is also important.

Though the development of different models is important the adoption in clinical practice is to be discussed with more details including certification of software, including it into hardware by vendors, etc. 

Minor issues

Section 2 lines 81-95 lacks citations

Several abbreviations were defined multiple times, for example RPA, AI, XAI, etc.

line 405 "There then must then then have concerns"?

Reviewer 4 Report

Comments and Suggestions for Authors
  1. The Abstract offers a helpful overview of the topic, but it would benefit from clearer structure and the inclusion of some key quantitative details. Adding one or two important statistics or specific tools or models mentioned in the review could help highlight the significance and practical impact of your findings.
  2. The introduction effectively highlights the growing role of AI in dentistry, but it would be stronger with a clearer statement of the review’s objectives or research questions. It would also be helpful to outline the review process and define the specific goals. Including a review framework such as a simple diagram could improve clarity and help readers better understand how this review contributes to the field. So it’s recommended to clearly identify the objectives, steps, and framework of the review, ideally illustrated with a diagram at the end of Introduction part.
  3. In section 2.1. Diagnostic Imaging and Analysis, there are many useful examples, but it kind of just lists them without really going deeper. Maybe try grouping the applications, like by imaging type or what task they do, and then compare how well they perform, how useful they are in clinic, or whether they've actually been validated. That way, the reader can better see what really stands out. Also, adding a summary table would be strongly recommended to make the information clearer and easier to follow.
  4. Section 2.2. Predictive Analytics for Treatment Planning, talks about a lot of interesting applications, which is great, but it would be even stronger if it made a clearer distinction between what's still theoretical and what's actually being used in real clinics. Including a few real-world examples, like case studies or clinical trials, could really help show what’s working and how it’s being applied in practice. Adding a Table is recommended.
  5. In Section 2.3 on AI in Real-time Documentation and Workflow Automation, there’s a lot of good information, but it starts to feel a bit text-heavy. Summarizing the key AI tools and their applications in a table or diagram could make things much clearer and easier to digest. So, it’s definitely recommended to add a table in this part to help reduce repetition and improve readability.
  6. In section 4.1. Ethical and Legal Concerns, its better to include examples of ongoing regulatory efforts such as EU AI Act, and FDA's software for medical device guidelines, etc, to make the review more stronger.
  7. The conclusion explains the main points well, but it would be better if you add 3 or 4 future steps or research ideas. Also, try to say clearly what this review gives to the field or how it helps future work. That would make the ending stronger.
Comments on the Quality of English Language

The manuscript is generally understandable, however, the quality of English needs improvement for clarity and readability. There are several instances of awkward phrasing, repetitive expressions, and grammatical inconsistencies throughout the text. It is recommended that the authors seek assistance from the English editing service to improve the overall flow, precision, and tone of the writing.

Round 2

Reviewer 3 Report

Comments and Suggestions for Authors

The authors addressed all my comments so I don't have any further suggestions, except one doubt: in the central box in Figure 2 is it "AL" or "AI" - capital "l" or capital "i".

Reviewer 4 Report

Comments and Suggestions for Authors

Thank you for your detailed and thoughtful responses to the previous review. You have addressed all the major concerns comprehensively, and the revised manuscript reflects these improvements well. I have no further comments and recommend the manuscript for publication.